# Development of a Children’s Visual Function Quality of Life (CVF-QoL) Instrument for Rural School-Going Children with Visual Impairments Within a South African Context: Item Generation and Validation

**DOI:** 10.3390/diagnostics15030331

**Published:** 2025-01-30

**Authors:** Tshubelela Sello Simon Magakwe, Rekha Hansraj, Zamadonda Nokuthula Xulu-Kasaba

**Affiliations:** Discipline of Optometry, University of KwaZulu Natal, Durban 4001, South Africa; hansrajr@ukzn.ac.za (R.H.); or 218074043@ukzn.ac.za (Z.N.X.-K.)

**Keywords:** vision-related quality of life, visual impairment, children, questionnaire, instrument, child-centered approach, self-report, interviews, validity

## Abstract

**Background**: Assessing visual function and quality of life in children with visual impairments is crucial for understanding its impact and evaluating interventions. While tools exist for developed countries, there is a lack of instruments for rural school-going children in middle- to low-income countries. This study aimed to develop and validate an instrument to measure Children’s Visual Function Quality of Life (CVF-QoL) for South African children aged 6 to 17 with uncorrected refractive errors, visual impairments, or blindness. **Methods**: The CVF-QoL instrument was created through a literature review and focus group discussions with experts and children. Readability was assessed, resulting in two versions of the CVF-QoL instrument. The contents of both versions were validated by experts, and pre-testing was performed with school children aged 6–17 years. **Results**: From the literature review, 769 items were identified, and 477 statements were generated from focus group discussions. After merging and eliminating duplicates, 91 items were classified into eight domains. The tool was divided into Version 1.1 for ages 10–17 (89 items) and Version 2.1 for ages 6–9 (63 items), both demonstrating strong clarity, coherence, and relevance. **Conclusions**: The CVF-QoL instruments are valid for evaluating the quality of life-related to visual function among rural school-going children with visual impairments in South Africa. This instrument thus provides an additional QoL tool to those already existing that may be more appropriate for measuring quality of life for rural school-going children in low- and middle-income countries.

## 1. Introduction

Children with early-onset, irreversible, severe vision impairments (VIs) may experience delays in motor, language, emotional, social, and cognitive development, with implications that can last a lifetime [1]. Furthermore, school-age children with VIs may also face lower levels of educational achievement [2]. It thus comes as no surprise that VI significantly impacts the quality of life of these individuals [3]. Quality of life (QoL) is defined by the World Health Organization (WHO) as “the perception an individual has of their position in life in the context of the culture and values in which they live and concerning their goals, expectations, standards, and concerns” [4]. Therefore, the vision-related QoL can be defined in the context of vision [5]. Essentially, vision-related QoL is the degree to which vision impacts how an individual can carry out daily living activities as they relate to their socioeconomic, emotional, and social well-being [6]. Increasingly, health services are focusing on capturing patient perceptions of their vision status [7]. Even in ophthalmic research, QoL measurement is being recognized as a key supplemental indicator [8].

The assessment of pediatric vision impairments has historically depended on clinical evaluations conducted by eye care practitioners, supplemented by observational reports from parental figures and educators [9]. This approach emerged as a practical necessity, given children’s inherent difficulties in articulating their visual challenges [10]. Nonetheless, the extent to which conventional clinical metrics—including visual acuity assessments, contrast sensitivity measurements, and visual field examinations—can effectively capture the pediatric patient’s lived experiences remains a debatable matter [9]. Recent studies suggest that quality-of-life instruments offer a promising alternative, providing a relatively economical yet reliable methodology for evaluating vision-related impacts on school-aged children [11]. These self-reporting mechanisms demonstrate advantages over proxy-based assessments in capturing the child’s direct perspective. Moreover, these tools can help measure the effectiveness of interventions and rehabilitation programs [12]. These evaluations are important for understanding the long-term impact of VIs on children’s development, education, and independence, and for guiding appropriate support and treatment strategies [13,14].

Evidence on various existing validated quality of life (QoL) assessment instruments designed for pediatric populations has been identified [2,10,11,14,15,16,17,18,19,20,21,22,23,24,25,26,27,28,29]. The National Eye Institute Visual Function Questionnaire (NEI-VFQ) has been effectively used to measure the quality of life (QoL) in individuals with vision impairments, but it is less suitable for children. This is because it includes irrelevant items like driving and omits critical aspects for child development, such as seeing from the blackboard or color perception [30]. The Cardiff Visual Ability Questionnaire for Children (CVAQC) targets children aged 5–18, developed in the UK based on focus groups of children with and without visual impairments. However, its applicability in developing countries is limited since all participants were white British, making it less relevant for South Africa’s predominantly Black population [2]. The Vision-Related Quality of Life Instrument for Young Children, developed in Dallas for children under seven, relies on proxy responses and does not specify suitable populations. It may be less effective than self-report instruments, which better capture children’s opinions [18].

The Impact of Visual Impairment on Children Questionnaire (IVI-C) is designed for children aged 8–18 in Australia and aims to measure vision-related QoL with positively framed items. Limitations include a small validation sample and difficulties in detecting changes in high-ability learners, necessitating further research with larger, randomized samples [23]. An instrument for children aged 8–18 with Juvenile Idiopathic Arthritis-Associated Uveitis measures vision-related QoL but may not apply to other conditions. Its validation is inconclusive due to issues like a small sample size, half the children having normal vision, and method limitations in measuring visual acuity [9]. Lastly, the LV Prasad Functional Vision Questionnaire, aimed at visually impaired children in India and developing countries, presents limitations such as potential bias from self-reporting, an unrepresentative sample, and contextual influences on content. Enhancements include adding mobility-related items and conducting further studies with a more diverse sample [10].

A significant limitation on existing tools emerges; approximately 85% of these instruments were developed within the context of developed economies. Consequently, they incorporate elements like driving and traffic sign comprehension, which hold minimal relevance for children in low- and middle-income countries. Activities central to these children’s daily experiences include navigating unpaved roads, engaging in sports on uneven terrain, and studying under suboptimal lighting conditions. The socioeconomic and cultural disparities between developed nations and African countries raise substantial concerns about these instruments’ reliability when applied to African children. This methodological constraint is further compounded by the instruments’ focus on developed-world leisure activities such as reading and digital entertainment, which stand in stark contrast to the typical rural African child’s daily activities such as agricultural labor, animal husbandry, outdoor recreational pursuits, and water collection from natural sources.

While the highest prevalence of children living with VIs is found within middle- to low-income countries [31,32,33,34], only four instruments have been designed using children in developing countries, three from India [10,17,25], and one from Saudi Arabia [22]. The recent scoping review highlighted the lack of an instrument specifically designed to measure visual function quality of life in children from rural Africa, South Africa to be specific. This makes it questionable to use some of the existing instruments without first modifying them for relevancy. The aim of this study is therefore to identify the domains, generate items, and validate and pre-test the content of the CVF-QoL instrument with a specific focus on South African rural school-going children.

## 2. Materials and Methods

The research followed the guidelines of the Declaration of Helsinki for research involving human subjects and received ethical approval from the Biomedical Research Ethical Committee of the University of KwaZulu-Natal (BREC/00003939/2022) and the Limpopo Provincial Review Ethics Committee (Ref:2/2/2). The development of CVF-QoL involved the following steps:

### 2.1. Step 1: Identification of Domain and Item Generation

#### 2.1.1. Domain Identification

The domains examined in this study were taken from the literature review. Before this part of the study, a scoping review was conducted on vision-specific quality of life (QoL) tools. A total of 20 tools were reviewed, and their items and domains were extracted for this part of the study [35]. This was performed to improve understanding of the phenomenon being studied, establish the boundaries of the domain, and facilitate the process of generating items and validating content [36]. Domains were selected based on the aims and objectives of the tools to be developed.

#### 2.1.2. Items Generation

To identify appropriate questions or items for the CVF-QoL instrument to fit the identified domains, the study employed deductive and inductive methods.

#### 2.1.3. Deductive Method

Initially, items were derived from a comprehensive literature review that systematically examined the availability and utilization of vision-specific QoL instruments for assessing VIs in children and adolescents [35]. A total of 20 instruments were included in this review and the scoping review paper on these instruments has been published elsewhere and can be accessed via this link, https://www.mdpi.com/2893610 accessed on 4 August 2024 [35].

#### 2.1.4. Inductive Methods

This part of the study utilized exploratory research methodology involving focus group discussions (FGDs) with two distinct participant groups: experts and children.

#### 2.1.5. Focus Group Discussions with Experts

Two FGDs were conducted with a panel of experts consisting of nine participants who worked with children regularly. A purposive sampling method was used to recruit the participants. Their experience ranged from 3 to 19 years at the time of the discussions and included teachers, occupational therapists, optometrists, and nurses. The FGDs were facilitated via Microsoft Teams, chosen as the most convenient method for all participants. Prior to the discussions, the participants were provided with an information sheet outlining the study and invited to participate. After signing the consent form, they received a Microsoft Teams meeting link through personal or work emails, detailing the date and time of the meeting. The discussions involved prompt questions about the role of vision in children’s lives and were recorded and transcribed. Each meeting with the experts lasted between 75 and 90 min. The statements related to vision were then categorized based on the domains adopted from the scoping review and were converted into items. The detailed report of this focus group discussion has been published and can be accessed via the link, https://doi.org/10.12688/f1000research.159464.1 accessed on 4 August 2024 [37], and its data can be accessed directly from the corresponding author via the email address.

#### 2.1.6. Focus Group Discussion with Children

A total of seven FGDs were conducted with children to gather their perspectives on the importance of vision in their daily lives. Forty-nine children aged 6–17 years from three schools in the Sekhukhune district of Limpopo province, South Africa, participated in this part of the study. Convenience sampling was used to sample the selected schools, and any learner aged 6–17 who volunteered to participate was allowed, as long as they did not have any other disabilities except for vision. Participants were categorized based on age, gender, and visual status, and all were of African ethnicity, residing in an area categorized as rural. The FGDs were held within the school premises, in classrooms provided by the schools. Parents were required to sign consent forms for their children to participate, and the children also signed assent forms if they wished to take part. One of the teachers was trained to assist the children to ensure their comfort during the discussions. Vision screening, which included visual acuity testing, visual field assessments, fundus examinations, and auto refraction, was conducted a few days before the FGDs to assess the participants’ visual status. Each meeting with children lasted between 60 and 80 min.

The discussions, for both focus groups, were guided by the prompt questions listed below.

What role does vision play in children’s lives overall?What school activities require children to have good vision?What types of recreational activities that children engage in require good eyesight?How does a child’s vision affect the way they socialize?If a child has vision problems, how would this impact their life and development?

The discussions were recorded and transcribed, allowing for the generation of statements. Similarly to the focus group discussions (FGDs) with experts, the comments were categorized into domains based on the literature, from which various items were derived. For more details on the data analysis methods used in these focus group discussions, please refer to the already published paper available via the following link: https://doi.org/10.12688/f1000research.159464.1 accessed on 4 August 2024 [37]. The information obtained from the literature review, FGDs with experts, and FGDs with children was carefully reviewed to identify any repeated items that could lead to redundancy. Several items were merged to avoid repetition with qualifying items used to create the first draft of the CVF-QoL instrument. Finally, the rating scales identified in the literature were reviewed, and suitable ones were adapted to be used as the rating scales for the CVF-QoL instrument.

### 2.2. Step 2: Content Validation of CVF-QoL Using a Panel of Experts

The initial drafts of the CVF-QoL Version 1.1 and Version 2.1 instruments were distributed to a group of experts considered so due to their knowledge in pediatric vision, visual disability, and/or tool development. Seven experts who were purposively sampled participated in this part of the study. Among them were three low-vision senior lecturers, two pediatrics lecturers, two optometrists with experience in QoL tools, and two educators. Firstly, the experts had to rate the level of clarity for each item on a scale of 1 to 4, with 4 being the item is clear and understandable, with good word order. Extra space was provided to comment on items or suggest revisions on items deemed unclear. Secondly, they had to rate the level of coherence of the item in measuring the aligned domain of the CVF-QoL instrument on a scale of 1 to 4, with 4 being the item is completely related to the domain it is measuring. Space was provided for them to comment on the item or suggest revisions. Thirdly, they had to rate the relevance of the item in measuring the domain of the CVF-QoL instrument on a scale of 1 to 4, with 4 being the item is very relevant and should be included. Space was provided for them to comment on the item or suggest revisions. Lastly, they had to comment on the overall instruments from the introduction to the last item.

The study used a statistical procedure called a content validity ratio to measure their consensus based on relevancy results. After collecting and considering the data and recommendations from the experts, the tool was modified and was ready for the pre-testing step.

### 2.3. Step 3: Pre-Testing

This step aimed to ensure that the questions aligned with the study’s subject and met the necessary criteria utilizing the cognitive interview technique. The initial questionnaire was reviewed by a group of learners conveniently sampled from two schools in Limpopo’s Sekhukhune district to assess the clarity, structure, and relevance of the questionnaires. The learners were of African ethnicity, aged 6 to 17 years, from rural areas. Additionally, they were categorized based on age, gender, and visual status.

This process aimed to confirm that the questions produced the intended data, identify and improve any questions that were confusing to children, resolve problematic or difficult-to-answer questions, ensure appropriate response options, shed light on how participants approached the topics, and identify any issues with question order. Additionally, questions were removed if fewer than 70% of the students agreed that they were relevant to them. The questionnaire and the rating were presented in English by the principal investigator and field workers who were fluent in both English and Sepedi to assist with interpretation if needed.

## 3. Results

A total of eight domains were adopted based on the literature review which are the “educational domain” which includes items that show how vision problems can affect learning. The “mobility and orientation domain” illustrates the challenges visually impaired children face when navigating new environments and the potential risks and difficulties they encounter due to poor vision. The “daily living skill domain” covers items related to difficulty with daily self-care tasks caused by poor vision. The “hobbies, leisure, and sport” section contains items that highlight how poor vision can impact recreational activities, which can hinder learning as children learn through play. The “social interaction domain” showcases the challenges visually impaired children face in developing bonds with family and friends, being excluded from many activities due to poor vision, and spending most of their time alone. The “psychological and emotional function domain” includes items that emphasize the impact of poor vision on confidence, self-esteem, and overall mental health. The “treatment domain” includes items that highlight the stigmas, strains, and costs associated with using eye drops or optical corrections for the management of VIs. Additionally, it emphasizes the financial burden that families experience when seeking proper treatment for their visually impaired children. Lastly, the “sociocultural domain” includes an item that highlights how the community profoundly influences the emotions, attitudes, values, beliefs, and interactions of its members regarding VIs.

Twenty resources were systematically reviewed, resulting in 749 extracted items. These resources consisted of instruments related to vision, designed to measure visual abilities, visual function, symptoms, or vision-related QoL in children with VIs [35]. The demographics of experts who participated in the FGDs are listed in Appendix A, while the children who participated can be found in Appendix B, located below the reference list. Focus group discussions generated 477 statements and comments, and can be accessed via this link: https://zenodo.org/records/13999156 accessed on 13 October 2024. The statements were considered for relevancy in the eight domains that were determined upon and included as was found appropriate.

After conducting a comprehensive review, removing duplicates, and consolidating items, the initial version of the CVF-QoL was developed. This version included rating scales that were agreed upon by the principal investigator and two research supervisors. The questions were assessed using the WebFX readability checker https://www.webfx.com/tools/read-able/ accessed on 3 July 2024 to determine the reading and comprehension level suitable for children. This process divided the CVF-QoL instrument into two versions: one for 6–9-year-olds with 66 items and another for 10–17-year-olds with 89 items.

### 3.1. Content Validation with Expert’s Results

Seven experts evaluated the two versions of the CVF-QoL instrument and found them to be coherent, comprehensive, and relevant. Table 1 below presents the relevancy scores for both versions of the CVF-QoL instrument, including the number of experts in agreement (Exp-A), the item content validity index (I-CVI), universal agreement (UA), the average score content validity index (S-CVI), the proportion relevance (PR), and the average score content validity universal average (S-CVI UA). The cut-off values for determining the content validity index (CVI) are as follows: CVI = 1 is considered valid for three to five experts, CVI ≥ 0.83 for six to eight experts, and CVI ≥ 0.78 for nine or more experts [38]. The S-CVI UA for the CVF-QoL version for ages 10–17 was 0.86, while the S-CVI UA for the CVF-QoL version for ages 6–9 was 0.85. Based on these calculations, we can conclude that the S-CVI average based on the item content validity index (I-CVI), the S-CVI average based on the PR, and the S-CVI UA all meet a satisfactory level. Therefore, both versions of the CVF-QoL instrument have achieved an acceptable level of content validity.

Table 2 below highlights some of the recommendations and modifications made on the items of both versions of the CVF-QoL instrument.

### 3.2. Pre-Testing Phase Results

After the content validation stage was completed, both versions of the CVF-QoL were modified as required and pre-tested with 30 learners. Fifteen learners aged 10 to 17 years were asked to evaluate the clarity, structure, and relevance of the entire questionnaire, as well as each item in the CVF-QoL Version 1.1. Meanwhile, another group of 15 learners aged 6 to 9 years rated the CVF-QoL Version 2.1. This process resulted in the reduction in some items, the modification of others, and the addition of new items. Table 3 presents the items that were retained for both versions of the CVF-QoL instrument. The time required to administer the questionnaires ranged from 15 to 20 min.

### 3.3. Rating Scales and Scoring of CVF-QoL

The rating scales and scoring for all items are detailed in Table 4. For example, the question “Indicate how much difficulty you have with the following activities due to vision problems” can be rated as follows:3 (very difficult) with a score of 0;2 (A little bit difficult) with a score of 50;1 (not difficult) with a score of 100;0 (does not apply to me) which is scored as “#” to indicate a missing response; this score should not be included when calculating the average scores.

### 3.4. Composite Score Calculation

To calculate an overall composite score for the CVF-QoL instrument, the scores from the vision-targeted sub-scales are averaged, i.e., we average the sub-scale scores rather than the individual items. By doing so, each sub-scale has an equal weight in the overall score, while averaging the individual items would give greater influence to the sub-scales with more items.

Formula:

Mean = ((Score for each item with a non-missing answer))/((Total number of items with non-missing answers))

Example:

With responses converted, ((25 + 100 + 25))/3 = 50

A score closer to 0 indicates poor QoL while a score closer to 100 indicates good QoL as indicated in Table 5.

## 4. Discussion

This study aimed to address the gap in vision-related quality of life instruments for rural school-going children in low–middle income countries. Scales are representations of latent constructs designed to measure behaviors, attitudes, and hypothetical scenarios that we anticipate may arise based on our theoretical understanding of the world [36]. However, it is important to note that these constructs cannot be directly assessed. While several scales have been developed to measure vision-related quality of life, there has been limited research specifically focused on rural school-going children from middle- to low-income communities. To address these gaps, two versions of a visual function quality of life (CVF-QoL) instrument for school-age children in rural South Africa were developed using a rigorous questionnaire development method. The CVF-QoL Version 1.2, intended for children aged 10 to 17 years, includes 89 items that assess eight different domains and is designed for older children. In contrast, the CVF-QoL Version 2.1, aimed at children aged 6 to 9 years, consists of 63 items that also evaluate eight domains, making it appropriate for younger children.

This study followed three key steps recommended by best practices for developing and validating scales in health, social, and behavioral research: domain and item generation, content validation, and pre-testing questions [36]. First, the domains and some items were adopted from the existing literature and modified as necessary. Several tools were developed in a similar manner [38,39,40]. Next, an inductive method of data collection was employed to identify items directly from the affected individuals. This approach was undertaken to capture the specific visual needs of the current population, as such an effort had not been made before. As a result, this study was able to identify an important and unique domain for these population studies, termed the sociocultural domain. This domain encompasses how cultural, historical, and social contexts shape perceptions of visual impairment (VI), how affected individuals are treated, and how VI is managed. It includes significant aspects, such as the association of VI with witchcraft or curses.

Both versions of the CVF-QoL instrument have demonstrated a satisfactory level of content validity. The Scores Content Validity Index Universal (S-CVI UA) for CVF-QoL Version 1.1 is 0.85, and for Version 2.1, it is 0.86. Both scores exceed the accepted cut-off score of 0.83 [21,41], which is applicable for a panel of six to eight experts, with seven experts having participated in this evaluation. After addressing the expert recommendations, the content of the instrument was deemed clear, comprehensive, and relevant for effectively measuring visual function QoL. To ensure the questions and answers are meaningful to the targeted population, both versions of the CVF-QoL instrument were administered to 30 learners, with fifteen assigned to each version. Angeles-Han (2011) utilized a similar method involving 13 learners for this step of their study when developing the vision-related quality of life instrument for children aged 8–18 years with juvenile idiopathic arthritis-associated uveitis [9]. The CVF-QoL instrument consists of an average of 152 items distributed across eight domains that assess the QoL related to visual function. It utilizes a four-point Likert scale to measure the level of difficulty, the level of agreement, and the frequency of issues, along with a three-point Likert scale for agreement in Version 2.1 as indicated in Table 4. The scoring method, domain averaging, and score interpretation are outlined in Table 5, adopted from the National Eye Institute-Visual Function Questionnaire (NEI VFQ-25) manual which is one of the most widely used instruments to assess the visual function quality of life [30].

The CVF-QoL instrument does assess the quality of life in children with visual impairment, and even though some of the questions (those dealing with specific activities) are relevant in a South African context, the majority of the questions should apply in any context as it focuses primarily on the impact of visual impairment on activities that children carry out, in general. While it may be most appropriate for low–middle income contexts where this type of rural setting is most likely to be found, the process followed in the development of this tool can be utilized in different contexts outside of South Africa in developing similar tools for different populations of concern, tailoring them for culturally appropriate activities and modifying them as required.

This study has the following limitations: 1. All the experts involved in developing the tool were local, focusing on the concerns of children in the area. The inclusion of international experts might have provided additional insights. 2. The children included in the development process were sourced from a single area in Limpopo, which may not represent all children in South Africa, especially those in rural regions. Despite these limitations, the CVF-QoL instrument is a valuable tool, as it appears to be the only quality of life (QoL) instrument for visually impaired individuals developed in an African and rural context.

## 5. Conclusions

The CVF-QoL instruments are valuable tools for evaluating the quality of life related to visual function in rural school-aged children with visual impairments (VIs). Particularly pertinent in South Africa, these instruments tackle the specific challenges faced by this population. They offer insights into how VI influences daily living and overall well-being, and they can also monitor changes in the quality of life following interventions, thereby aiding the assessment of management strategies. This paper focuses on the generation and validation of items for a CVF-QoL tool designed to assess the impact of visual impairment in rural school children. Further validation of the final tool, including an examination of its reliability, will be presented in a subsequent paper.

## Figures and Tables

**Table 1 diagnostics-15-00331-t001:** Expert’s scoring on the relevancy of both CVF-QoL instruments.

CVF-QoL Version 1.1								CVF-QoL Version 2.1							
Item (Domain)	Number of Experts in Agreement	I-CVI	UA	Item (Domain)	Number of Experts in Agreement	I-CVI	UA	Item (Domain)	Number of Experts in Agreement	I-CVI	UA	**Item (Domain)**	**Number of Experts in Agreement**	**I-CVI**	**UA**
1 ^[SL]^	7	1	1	46 ^[HLS]^	7	1	1	1 ^[SL]^	7	1	1	46 ^[PEF]^	7	1	1
2 ^[SL]^	7	1	1	47 ^[HLS]^	7	1	1	2 ^[SL]^	7	1	1	47 ^[PEF]^	7	1	1
3 ^[SL]^	6	0.86	0	48 ^[HLS]^	7	1	1	3 ^[SL]^	7	1	1	48 ^[PEF]^	7	1	1
4 ^[SL]^	7	1	1	49 ^[SI]^	7	0.86	0	4 ^[SL]^	7	1	1	49 ^[PEF]^	7	1	1
5 ^[SL]^	7	1	1	50 ^[SI]^	7	1	1	5 ^[SL]^	7	1	1	50 ^[PEF]^	6	0.86	0
6 ^[SL]^	7	1	1	51 ^[SI]^	7	1	1	6 ^[SL]^	7	1	1	51 ^[PEF]^	5	0.71	0
7 ^[SL]^	4	0.57	0	52 ^[SI]^	7	1	1	7 ^[SL]^	4	0.57	0	52 ^[PEF]^	6	0.86	0
8 ^[SL]^	7	1	1	53 ^[SI]^	7	1	1	8 ^[SL]^	7	1	1	53 ^[T]^	7	1	1
9 ^[SL]^	7	1	1	54 ^[PEF]^	7	1	1	9 ^[SL]^	7	1	1	54 ^[T]^	7	1	1
10 ^[SL]^	7	1	1	55 ^[PEF]^	7	1	1	10 ^[SL]^	7	1	1	55 ^[T]^	7	1	1
11 ^[SL]^	7	1	1	56 ^[PEF]^	7	1	1	11 ^[SL]^	6	0.86	0	56 ^[T]^	7	1	1
12 ^[SL]^	7	1	1	57 ^[PEF]^	7	1	1	12 ^[MO]^	7	1	1	57 ^[T]^	7	1	1
13 ^[SL]^	7	1	1	58 ^[PEF]^	7	1	1	13 ^[MO]^	7	1	1	58 ^[T]^	7	1	1
14 ^[SL]^	7	1	1	59 ^[PEF]^	7	1	1	14 ^[MO]^	7	1	1	59 ^[T]^	7	1	1
15 ^[SL]^	7	1	1	60 ^[PEF]^	7	1	1	15 ^[MO]^	7	1	1	60 ^[T]^	7	1	1
16 ^[SL]^	7	1	1	61 ^[PEF]^	7	1	1	16 ^[MO]^	6	0.86	0	61 ^[SC]^	7	1	1
17 ^[MO]^	7	1	1	62 ^[PEF]^	7	1	1	17 ^[MO]^	7	1	1	62 ^[SC]^	7	1	1
18 ^[MO]^	7	1	1	63 ^[PEF]^	7	1	1	18 ^[DLS]^	7	1	1	63 ^[SC]^	7	1	1
19 ^[MO]^	7	1	1	64 ^[PEF]^	7	1	1	19 ^[DLS]^	7	1	1	64 ^[SC]^	7	1	1
20 ^[MO]^	7	1	1	65 ^[PEF]^	7	1	1	20 ^[DLS]^	6	0.86	0	65 ^[SC]^	7	1	1
21 ^[MO]^	7	1	1	66 ^[PEF]^	7	1	1	21 ^[DLS]^	7	1	1	66 ^[SC]^	7	1	1
22 ^[MO]^	6	0.86	0	67 ^[PEF]^	7	1	1	22 ^[DLS]^	7	1	1				
23 ^[MO]^	7	1	1	68 ^[PEF]^	7	1	1	23 ^[DLS]^	7	1	1				
24 ^[MO]^	7	1	1	69 ^[PEF]^	7	1	1	24 ^[DLS]^	7	1	1				
25 ^[DLS]^	7	1	1	70 ^[PEF]^	7	1	1	25 ^[DLS]^	7	1	1				
26 ^[DLS]^	7	1	1	71 ^[PEF]^	7	1	1	26 ^[DLS]^	7	1	1				
27 ^[DLS]^	7	1	1	72 ^[PEF]^	6	0.86	0	27 ^[DLS]^	6	0.86	0				
28 ^[DLS]^	7	1	1	73 ^[PEF]^	7	1	1	28 ^[DLS]^	6	0.86	0				
29 ^[DLS]^	7	1	1	74 ^[T]^	6	0.86	0	29 ^[HLS]^	7	1	1				
30 ^[DLS]^	7	1	1	75 ^[T]^	6	0.86	0	30 ^[HLS]^	7	1	1				
31 ^[DLS]^	7	1	1	76 ^[T]^	7	1	1	31 ^[HLS]^	7	1	1				
32 ^[DLS]^	7	1	1	77 ^[T]^	6	0.86	0	32 ^[HLS]^	7	1	1				
33 ^[DLS]^	6	0.86	0	78 ^[T]^	7	1	1	33 ^[HLS]^	7	1	1				
34 ^[DLS]^	7	1	1	79 ^[T]^	7	1	1	34 ^[HLS]^	7	1	1				
35 ^[DLS]^	7	1	1	80 ^[T]^	6	0.86	0	35 ^[SI]^	5	0.71	0				
36 ^[DLS]^	7	1	1	81 ^[T]^	7	1	1	36 ^[SI]^	7	1	1				
37 ^[DLS]^	7	1	1	82 ^[T]^	7	1	1	37 ^[SI]^	7	1	1				
38 ^[DLS]^	7	1	1	83 ^[SC]^	7	1	1	38 ^[SI]^	7	1	1				
39 ^[DLS]^	7	1	1	84 ^[SC]^	7	1	1	39 ^[PEF]^	7	1	1				
40 ^[DLS]^	7	1	1	85 ^[SC]^	7	1	1	40 ^[PEF]^	7	1	1				
41 ^[DLS]^	7	1	1	86 ^[SC]^	7	1	1	41 ^[PEF]^	7	1	1				
42 ^[DLS]^	7	1	1	87 ^[SC]^	7	1	1	42 ^[PEF]^	7	1	1				
43 ^[HLS]^	7	1	1	88 ^[SC]^	5	0.71	0	43 ^[PEF]^	7	1	1				
44 ^[HLS]^	7	1	1	89 ^[SC]^	6	0.86	0	44 ^[PEF]^	7	1	1				
45 ^[HLS]^	7	1	1	90 ^[SC]^	6	0.86	0	45 ^[PEF]^	7	1	1				
S-CVI Ave = 0.97								S-CVI Ave = 0.97							
S-CVI Ave (based on PR) = 0.97								S-CVI Ave (based on PR) = 0.95							
S-CVI UA = 0.85								S-CVI UA = 0.86							

[SL] = School and Learning; [MO]= Mobility and Orientation; [DLS]= Daily Living Skills; [HLS]= Hobbies Leisure and Sport; [SI] = Social Interaction; [PEF] = Psychological and Emotional Function; [T] = Treatment; [SC] = Sociocultural; I-CVI = Item-Content Validity Index; UA = Universal Agreement; PR = Proportion Relevance; S-CVI Ave = Score-Content Validity Index Average; S-CVI Ave (based on PR) = Score-Content Validity Index based on Proportional Relevancy; S-CVI UA = Score Content Validity Universal Average.

**Table 2 diagnostics-15-00331-t002:** Experts’ comments and items modified.

CVF-QoL Version 1.1			CVF-QoL Version 2.1		
Original Item	Experts Comments	Revised Item	Original Item	Experts Comments	Revised Item
Identifying colors (e.g., while coloring)	Maybe consider seeing colors, instead of identifying.	Matching colors (e.g., while coloring or painting)	Completing my school work on time	Suggestion: Use “finishing” instead of “completing”.	Finishing my schoolwork on time
Remembering what I have read	I think it should be rephrased or removed. Unless you want to test something on seeing and memory.	Deleted	Remembering what I have read	I think it should be rephrased or removed. Unless you want to test something on seeing and memory.	Deleted
I do not do as well in my tests and exams as others	Remember that there are over-achievers who may answer or agree to this despite no visual problems. Maybe consider ‘I do not do as well in my tests and exams as others, because of my eyes’.	I do not do as well in my tests and exams as others, because of my eyesight	I cannot see well from the chalkboard	Duplicate, same as no.1. To be deleted.	Deleted
Telling the time on a wall clock	If this question is linked to ascertaining visual function for a distance task, consider seeing who is at the door.	Seeing the time on a wall clock	I experience double vision when I read	Simplify it for this age group, they might not understand the question.	Words are becoming double when I read
Introduction of new items	Add items related to children’s hobbies under the domains Hobbie, Leisure, and Sport.	1. It is difficult to read the story books because of my eyesight. 2. It is difficult to create crafts with clay or boxes because of my eyesight. 3. I find it difficult to do knitting because of my eyesight.	My head is sore when I read	I suggest you rephrase it to replace sore head with headaches.	I sometimes experience headaches when I read
I feel different from other children because my eyes go in and out	Consider left and right or side to side. In and out seems inclined to scientific optometry jargon.	I feel different from other children because my eyes are crossed	I skip lines when I read or write	Please make this text easier to understand for this age group.	Reading or writing along a line in a book
I get bullied at school because of wearing glasses	This item fits well under the SocCultural domain.	The item was moved from the Psychology and emotional function to the SocioCulturak function.	Telling the time on a wall clock	If this question is linked to ascertaining visual function for a distance task, consider seeing the time on a wall clock.	Seeing the time on a wall clock
Introduction of new items	Recommended item under treatment domain.	1. I cannot see clearly even when I wear my glasses 2. The eyedrop is uncomfortable for my eyes	I don’t like the way I am	Maybe say ‘I don’t like the way I am because of my eyes’	I don’t like the way I am because of my eyesight
I am mistreated by other children and other village members because of my eye problems	This text must be removed. It is similar to item number 2.	Deleted	People think I am cleaver less because of my poor eyesight	Rather say people think I am not clever because I cannot see.	People think I am not clever because of my poor eyesight
My parents think I am cursed or bewitched	This can be combined with number 1.	Deleted	I do not feel good about myself because I cannot see well	This is the same as item number 6 and should be deleted.	Deleted
			I feel ashamed about attending a special school	Remove for this age group.	Deleted

**Table 3 diagnostics-15-00331-t003:** All versions of the CVF-QoL instrument.

Items for CVF-QoL Version 1.1 [Domain]			Items for CVF-QoL Version 2.1 [Domain]	
Indicate how much difficulty you have with the following activities because of vision problems:	33. Playing games requires me to see things in the near such as Morabaraba, Diketo, Mmela, etc. ^[HLS]^	65. I spend a lot of time on vision treatment (eye doctor appointments, patching, eye drops, and therapy) ^[T]^	How hard or easy is it for you to do the following things because of your eyesight problems:	32. I often fall when I walk because I cannot see well ^[MO]^
1. Seeing clearly what is written on the chalkboard ^[SL]^	34. Maintaining eye contact when talking with your friend ^[SI]^	66. I find it hard to clean my glasses ^[T]^	1. Seeing clearly what is written on the chalkboard ^[SL]^	33. I am not sure of how far a car or taxi is to me when I am crossing the road ^[MO]^
2. Drawing ^[SL]^	35. Taking part in various activities (like sporting events) with friends ^[SI]^	67. The eyedrop is uncomfortable for my eyes ^[T]^	2. Drawing ^[SL]^	34. I need someone to stay at home with me because I cannot see well ^[DLS]^
3. Matching colors (e.g., while coloring or painting) ^[SL]^	36. Making friends ^[SI]^	68. I cannot see clearly even when I wear my glasses ^[T]^	3. Matching colors (e.g., while coloring or painting) ^[SL]^	35. I often lose in most games because I cannot see ^[HLS]^
4. Using a pair of scissors to get the desired shapes ^[SL]^	Please indicate if these things happen to you:	69. My family members are ashamed of me because of my eyesight ^[SC]^	4. Using a pair of scissors to get the desired shapes ^[SL]^	36. It is difficult to read the story books because of my eyesight ^[HLS]^
5. Completing my school work on time ^[SL]^	37. My eyesight affects how fast I can read and write ^[SL]^	70. I feel like I am cursed for not seeing well ^[SC]^	5. Finishing my school work on time ^[SL]^	37. It is difficult to create crafts with clay or boxes because of my eyesight ^[HLS]^
6. Reading from books including my textbooks ^[SL]^	38. My eyes hurt and tear up when I read or do close work ^[SL]^	Indicate how often this happens:	6. Reading from books including my textbooks ^[SL]^	38. Other children don’t want to play with because of my eyesight ^[SI]^
7. Seeing pictures clearly in my book (e.g., Map book) ^[SL]^	39. Words become double when I read ^[SL]^	71. I am afraid of drowning when fetching water by the river ^[DLS]^	7. Copying correctly from the board or books ^[SL]^	39. It is hard to see if someone is happy, sad, or angry when they are talking to me ^[SI]^
8. Reading or writing along a line in a book ^[SL]^	40. I think I would benefit better from the special school for people who cannot see well ^[SL]^	72. I get closer to the television (TV) screen so that I can see what’s happening on TV ^[HLS]^	8. Reading or writing along a line in a book ^[SL]^	40. I don’t like going for my eye check-up ^[PEF]^
9. Using public transport (e.g., taxi or bus) ^[MO]^	41. I do not do as well in my tests and exams as others, because of my eyesight ^[SL]^	73. I am scared that I might go blind ^[PEF]^	9. Moving around by yourself without tripping into objects or people ^[MO]^	41. I feel different from other children because my eyes are crossed ^[PEF]^
10. Moving around by yourself without bumping into objects or people ^[MO]^	42. I struggle to read what other people have written because of my eyesight ^[SL]^	74. I am worried about my poor vision ^[PEF]^	10. Moving around safely by myself in places I don’t know ^[MO]^	42. Wearing glasses makes me feel different ^[PEF]^
11. Walking safely in new places without help ^[MO]^	43. I get headaches after reading for some time ^[SL]^	75. I get bullied at school because I wear glasses ^[SC]^	11. Walking home ^[MO]^	43. I feel sad when friends laugh at me because I cannot see well ^[PEF]^
12. Walking home ^[MO]^	44. I copy from other classmate’s notes because I struggle to see clearly on the board ^[SL]^	76. People think I am less intelligent because of my poor eyesight ^[PEF]^	12. Walking on bumpy ground ^[MO]^	44. I find it hard to clean my glasses ^[T]^
13. Walking on uneven ground ^[MO]^	45. I am not sure of how far a car or taxi is to me when I am crossing the road ^[MO]^	77. I am afraid to tell the teacher that I cannot see well in class ^[PEF]^	13. Seeing someone across the street when they are calling out to you ^[MO]^	45. I don’t like it when someone puts drops in my eyes ^[T]^
14. Seeing someone across the street when they are calling out to you ^[MO]^	46. I often fall when I walk because I cannot see well ^[MO]^	78. I am bothered because my parents tend to treat me differently as compared to my other siblings due to eyesight ^[PEF]^	14. Going to the toilet ^[DLS]^	46. I cannot see clearly even when I wear my glasses ^[T]^
15. Fetching water ^[DLS]^	47. I often need someone to stay at home with me because I cannot see well ^[DLS]^	79. I feel worthless because I cannot see well ^[PEF]^	15. Seeing the right money when at the shop ^[DLS]^	47. I do not like the way I look when wearing glasses ^[T]^
16. Going to the toilet ^[DLS]^	48. I struggle to look after my siblings when we are left home alone ^[DLS]^	80. It is hard for me to defend myself verbally, even if I am right ^[PEF]^	16. Seeing the time on a wall clock ^[DLS]^	48. My family is ashamed of me because of my poor eyesight ^[SC]^
17. Locking or unlocking the door ^[DLS]^	49. I often lose in most games due to my eyesight ^[HLS]^	81. I feel ashamed about attending a special school ^[PEF]^	17. Seeing the time on a wristwatch ^[DLS]^	49. My parents think I am under a curse or bewitched because of my eyesight ^[SC]^
18. Making out differences in coins or notes when I want to pay because of my eyesight ^[DSL]^	50. It is difficult to read the story books because of my eyesight ^[HLS]^	82. When I wear my glasses, I have to squint to see things ^[T]^	18. Taking care of myself (like washing my face, applying toothpaste, brushing my teeth, dressing myself, lacing shoes, etc.) ^[DLS]^	50. I feel like I am cursed for not seeing well ^[SC]^
19. Taking goats and cows to the field for grazing ^[DLS]^	51. It is difficult to create crafts with clay or boxes because of my eyesight ^[HLS]^	83. Glasses hurt my nose/ears ^[T]^	19. Pouring water/drink into a cup or glass ^[DLS]^	How many times does this happens to you:
20. Washing my school uniform so that it gets clean ^[DLS]^	52. I find it difficult to do knitting because of my eyesight ^[HLS]^	84. I feel dizzy/nausea when I wear my glasses ^[T]^	20. Seeing the food on my plate when eating ^[DLS]^	51. I go closer to the TV so that I can see better ^[HLS]^
21. Doing household chores (like sweeping, washing dishes, cooking, preparing a meal ^[DLS]^	53. People are getting frustrated with you because of your eyesight ^[SI]^	85. My glasses sometimes break or fall off while I am wearing them ^[T]^	21. Going to the tuckshop alone ^[DLS]^	52. I am scared that I might go blind ^[PEF]^
22. Seeing the time on a wall clock ^[DLS]^	54. Other children don’t want to play with me because I cannot see ^[SI]^	86. I have been told that I am cursed or bewitched for not being able to, see clearly ^[SC]^	22. Using the cell phone (for example to dial a number or read messages) ^[DLS]^	53. I am worried that I cannot see well ^[PEF]^
23. Seeing the time on a wristwatch ^[DLS]^	55. It is hard to see if someone is happy, sad, or angry when they are talking to me ^[SI]^	87. I am treated differently from other children in my village because of eye problems ^[SC]^	23. Taking part in athletics (running, jumping, throwing, and race walking) ^[HLS]^	54. I get bullied at school because I wear glasses ^[SC]^
24. Taking something out from a cupboard ^[DLS]^	56. I become stressed when I have to go for my eye check-up ^[PEF]^	88. Some people believe I cannot think properly because of my eye problem ^[SC]^	24. Playing ball games (soccer, tennis, Mokoloti, etc.) ^[HLS]^	55. People think I am not clever because of my poor eyesight ^[PEF]^
25. Taking care of myself (like washing my face, applying toothpaste, brushing my teeth, dressing myself, lacing shoes, etc.) ^[DLS]^	57. I feel different from other children because my eyes are crossed ^[PEF]^	89. Some people think that God is punishing my parents because of my eye problems ^[SC]^	25. Playing games requires me to see things in the distance such as Dithini, Thapo, and Wulu ^[HLS]^	56. I am afraid to tell the teacher that I cannot see well in class ^[PEF]^
26. Pouring water/drink into a cup or glass ^[DLS]^	58. Wearing glasses makes me feel different from other children ^[PEF]^		26. Playing games requires me to see things that are near such as Morabaraba, Diketo, Mmela, etc. [HLS]	57. I feel ashamed about attending a special school ^[PEF]^
27. Seeing the food on my plate while eating ^[DLS]^	59. I feel like people are impatient with me due to my eyesight ^[PEF]^		Please indicate if these things happen to you:	58. When I wear my glasses, I have to cross my eyes to see things ^[T]^
28. Going to the tuckshop alone ^[DLS]^	60. I feel sad because I may not be able to choose the career I want because of my eyesight ^[PEF]^		27. My eyes hurt when I read ^[SL]^	59. Glasses hurt my nose/ears ^[T]^
29. Using the cell phone (for example to dial a number or read messages) ^[DLS]^	61. I feel sad when I get corrected on things I did not see clearly ^[PEF]^		28. I cannot see what other people write in their books ^[SL]^	60. I feel dizzy when I wear my glasses ^[T]^
30. Taking part in athletics (running, jumping, throwing, and race walking) ^[HLS]^	62. I feel sad when friends laugh at me because I cannot see well ^[PEF]^		29. Words are becoming double when I read ^[SL]^	61. My glasses sometimes break or fall off while I am wearing them ^[T]^
31. Playing ball games (soccer, tennis, Mokoloti, etc.) ^[HLS]^	63. I feel like I am a burden to my family members and the people around me because of my eyesight ^[PEF]^		30. I have a sore head when I read ^[SL]^	62. I am being mistreated by other children in my village because of my eye problems ^[SC]^
32. Playing games requires me to see things in the distance such as Dithini, Thapo, and Wulu ^[HLS]^	64. I have low confidence in myself because of my poor eyesight ^[PEF]^		31. People are getting annoyed with me because of my eyesight ^[SL]^	63. I cannot see so people say I cannot think well ^[sc]^

[SL] = School and Learning; [MO] = Mobility and Orientation; [DLS] = Daily Living Skills; [HLS] = Hobbies Leisure and Sports; [SI] = Social Interaction; [PEF] = Psychological or Emotional Function; [T] = Treatment; [SC] = Sociocultural.

**Table 4 diagnostics-15-00331-t004:** Rating scales and values for CVF-QoL instrument.

Responds	Change the Original Response Category	To Record the Value of:
Indicate how much difficulty you have with the following activities because of vision problems		
Very difficult	3	0
A little bit difficult	2	50
Not difficult	1	100
Does not apply to me	0	# (missing)
Please indicate if these things happen to you		
Yes	3	0
Sometimes	2	50
No	1	100
Does not apply to me	0	# (missing)
Indicate how often this happens		
All the time	3	0
Sometimes	2	50
Never	1	100
Do not apply to me	0	# (missing)
How hard or easy is it for you to do the following things because of your eyesight problems		
Very hard for me	3	0
Not so easy for me	2	50
Very easy for me	1	100
I do not do this thing	0	# (missing)
Please indicate if these things happen to you		
Yes	2	0
No	1	100
Does not apply to me	0	# (missing)
How many times does this happens to you		
All the time	3	0
Sometimes	2	50
Never	1	100
Do not apply to me	0	# (missing)

# = missing data and should not be counted when averaging the domain. Scores represent the achieved percentage of the total possible score, e.g., a score of 50 represents 50% of the highest possible score.

**Table 5 diagnostics-15-00331-t005:** CVF-QoL instrument score indication.

Score Ranges	Indication
70–100	Good quality of life (QoL)
50–74	Fair QoL
25–49	Poor QoL
0–24	Very poor QoL

## Data Availability

The generated statements and comments from the focus group discussion were deposited online and can be accessed by this link: https://zenodo.org/records/13999156 accessed on 3 July 2024 [42]. However, the recording, transcription, codes, and all other data can be accessed by emailing the corresponding author.

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
