# Peer review of "Development of a Children’s Visual Function Quality of Life (CVF-QoL) Instrument for Rural School-Going Children with Visual Impairments Within a South African Context: Item Generation and Validation"

_diagnostics, 2025, doi:10.3390/diagnostics15030331_

Round 1
Reviewer 1 Report
Comments and Suggestions for Authors
The authors developed an instrument to measure the Quality of Life of Children in South Africa based on their visual function. This instrument is relevant to children living in rural areas and in a different cultural and economic context than those in urban areas, where most of the questionnaires were developed. The authors used a multi-staged approached through literature review, focus groups, and pre-testing to determine the validity of the questionnaire, which was shown to have a high validity. Though this may be a considerable tool, there are numerous general and specific comments to be answered.
General questions:
How could this be tested or applied to different contexts outside of South Africa?
What is the potential impact of this questionnaire (rehabilitation treatment, optometric assessments, ophthalmological treatment)?
Who should administer the questionnaire (teachers, optometrists, parents, etc.)?
Where should it be administered?
How should it be administered (guidelines)?
Have any of these children been followed up to determine if this was accurately measuring the vision component or possible other components, such as cognitive disabilities, hearing disabilities, communication disorders, etc.?
Is this intended to be focused on individuals with temporary vision problems or permanent, or both? This could inform/guide the administrator into the appropriate follow-up.
Specific questions/comments:
Introduction:
Though the other studies were performed in urban areas, it is unclear how well the other studies have performed in measuring QoL for children living in rural areas. Even if a questionnaire is designed in an urban area, it could potentially have high validity in rural areas and other contexts as well. It could be useful to know if studies specifically mentioned the issues highlighted the areas of concern raised by the authors, such as agriculture, collection from natural sources, etc. Though the point is valid, this does not directly relate to the lack of accuracy of the other instruments. The authors do not cite other articles in regards to this point.
How well-designed are the other instruments in detecting vision function, or vision-related QoL? Are there issues related to the non-specificity whereby it also picks up those with cognitive impairments or intellectual disabilities for instance? For instance, as reading is a highly cognitive process, questions related to reading may not always be vision-specific. There should be references to this point as well.
Methodology
There are several areas that appear to be missing/unclear in the methodology. Though some of the links clear up certain points for the reader, there are multiple areas that are difficult to comprehend. The following areas, specifically:
· -Recruitment methods are not disclosed
· -Inclusion/exclusion criteria are not mentioned
· -The time period and length estimated for participants during each stage
· -The questions that were asked of children, their parents, and professionals
· -The guidance that was given to the participants regarding the study (what were they requested to do?)
· -Method of transcription and analysis of the comments from all groups
· -Methods for reviewal of comments and subsequent categorization
· -Language of the administration of the questionnaire, responses, and comments
Additionally, there is some information in the methods that is better suited for the results, such as the experience of experts.
Results
A table for the participants and experts would greatly help to understand who was interviewed in the focus group and who participated in the pre-testing.
Also, the 477 statements was mentioned, but the quality or type of information that was received is not clear.
The 2nd sentence above Table 2 should be eliminated. It looks like a previous comment.
In Table 2, it is difficult to see which section the comments are referring to. It would be helpful to have borders to more easily read the table.
The length of administration of the questionnaire is not given.
An average number of questions is provided, but is this from averaging the 2 questionnaires or because some of the questions are conditional and thus not asked of certain children?
Discussion
It should start by referring to the reason for conducting the study mentioned in the introduction, rather than talking about scales.
There is little information about suggestions from the authors about the implications from this questionnaire and how they suggest it should be implemented (see general comments above).
Author Response
Please see the attachment in response to your reviews and comments.
Thank you

Reviewer 2 Report
Comments and Suggestions for Authors
1- In abstract briefly mention what your instrument can add to previous instruments for use in middle to low income countries.
2- In methodology section give more details about the process of choosing previous vision-specific QoL instruments for assessing VI in children. What exact parameters did you use for choosing these instruments?
3- Give more details about the panel of experts involved in constructing your instrument and their qualifications in methods section. Were they randomly selected? Did they have any previous expertise in working with children with vision disabilities?
4- Give more details about the criteria used for choosing the final questions used in the instrument.
5- Add a statistical methods section and give more details about the statistical process of testing your instrument for validity and reliability.
6- ٍPlease expand the discussion section and compare and contrast your instrument with some previous instruments from the literature regarding its probable advantages and limitations.
7- Add a limitations section to discussion and mention the limitations of your study.
8- Summarize your conclusion and avoid broad statements like "The CVF-QoL instruments are reliable tools designed to evaluate the quality of life 328 concerning visual function specifically for rural school-going children who experience VI." Conclusion should be a brief statement of your results and their usefulness.
Author Response
Please find the attachment in response to your reviews and comments
Thank you
Kind regards

Round 2
Reviewer 1 Report
Comments and Suggestions for Authors
I appreciate the response from the authors from several of the points in the previous review. There are considerable improvements from the previous version.
However, there remain questions regarding the structure and information provided in the manuscript. The information regarding the other tools was in the discussion, but is not mentioned in the introduction. As this is relevant to understand the importance of this research, the information should be represented in the introduction. The recrutement and sampling methods and reasons for choosing certain students are still relatively unclear. It appears that while some of the methodological questions were answered, they do not necessarily appear in the most recent version of the manuscript.
Author Response
Good day
Please see attachment
kind regards

Reviewer 2 Report
Comments and Suggestions for Authors
The manuscript has improved.
Author Response
Good day,
thank you for enhancing this manuscript; your input is invaluable.
kind regards